# Channel Deformations and Hazardous Processes of the Left-Bank Tributaries of The Angara River (Eastern Siberia)

**Marina Y. Opekunova** [1,*]**, Natalia V. Kichigina** [1]**, Artem A. Rybchenko** [2] **and Anton V. Silaev** [1]

1   V.B. Sochava Institute of Geography SB RAS, 664033 Irkutsk, Russia
2   Institute of the Earth's Crust SB RAS, 664033 Irkutsk, Russia
*   Correspondence: opek@mail.ru

**Abstract:** The influence of anthropogenic and natural factors in the trends and mechanisms of development at various topological levels is determined based on relevant information on the structure and dynamics of fluvial systems in the south of Eastern Siberia in various geodynamic settings. This article considers the current spatial and temporal dynamics of the hydrological conditions of the vast territory of the Angara River and its influence on channel deformations and the manifestation of dangerous processes. An analysis of fluctuations in the maximum runoff using differential integral curves resulted in the identification of six periods of water content according to the maximum annual discharges for the period spanning from the beginning of observations to 2020 for the rivers under consideration. The dynamics and intensity of manifestation of hydrological and geological hazardous processes are demonstrated using a series of studies conducted under various geodynamic conditions. Catastrophic floods brought on by enhanced cyclonic activity are accompanied by the destruction of the bank. The highest rate of bank erosion in the plains is 1.5 to 2 m per year, and for rivers in mountains and piedmonts it is 2 to 6 m per year. An analysis of the dynamics of the development of floodplain–channel complexes in the Upper Angara region makes it possible to distinguish two zones of actively developing floodplain–channel complexes: piedmont and estuarine, separated by a relatively stable plain zone.

**Keywords:** river basins; hydrological regime; flood; channel deformations; morphodynamics; hazardous processes; Eastern Siberia

## 1. Introduction

The study of the processes of fluvial relief formation, in particular, a comprehensive study of the formation and development of river valleys, reveals a number of features of the development of natural components, including the rhythm of exogenous relief formation [1–7], which is an important component in the process of transformation and evolution of landscapes [8–10]

Furthermore, such studies perform the functions of predicting and preventing the occurrence of environmental tension associated with hazardous hydrological phenomena, the dynamics of channel processes, and the transformation of valley landscapes manifested under anthropogenic pressure [11–13].

The area under study covers river basins with an area of more than 1.6 thousand km², occupying mountain and piedmont–plain areas. The lower reaches belong to areas of long-term development that are experiencing significant anthropogenic pressure, including those associated with the construction of a cascade of hydroelectric power stations on the Angara River [14,15].

The extreme hydrological and geological processes are characteristic of both mountainous [16–19] and lowland regions [20–23].

Recently, there has been an increase in cyclonic activity [24] and catastrophic rain floods, which are recognized as one of the leading factors in emergency situations in

the southern regions of Irkutsk Oblast [25,26]. In this regard, it is important to have an idea about the spatial and temporal differentiation and dynamics of the hydrological regime, as well as the degree of influence of extreme floods on the intensity of dangerous geological processes.

## 2. Objectives and Methods of the Study

The area under study is in the upper reaches of the Angara, including sections of the lower reaches of its left tributaries Kitoi, Irkut, Belaya, and Iya (Table 1).

**Table 1.** Morphometric characteristics of the river basins.

| No. | River | Basin Area, km² | River Length, km | Length of the River within the Plain–Platform Area, km |
|---|---|---|---|---|
| 1. | Irkut | 15,000 | 488 | 60 |
| 2. | Kitoi | 9190 | 316 | 94 |
| 3. | Belaya | 18,000 | 359 | 79 (from the watersmeet of Malaya and Bol'shayaBeaya) |
| 4. | Iya | 18,100 | 484 | 133 |

The specificity of the river valley formation of the left-bank tributaries of the Angara is primarily determined by their location at the junction of the orogenic and platform areas. The transverse, sublatitudinal-northeast depressions give a 'keyboard' character to the trough. The presence of differently oriented fault zones of different topological orders is another feature of the territory's structure and development that has influenced the morphology of the valleys and their parts. The bottoms of the valleys are filled with Neogene and Quaternary pebbles and sands.

The combination of geological and geomorphological structure and climatic conditions determines the heterogeneity of the area and the formation of geomorphological units of lower rank. Geomorphologically [27], the river basins are located within the stepped upwarp of the Eastern Sayan and the Central Siberian Plateau (Figure 1). Hydromorphological analysis was performed at three major geomorphological regions: mountains, piedmonts (Eastern Sayan), and plains (Irkutsk–Cheremkhovskaya Plain).

The distribution of intra-annual flow, water discharge, and sediment yield occurdue to the location of a significant part of the catchment area of the left-bank tributaries in the mountainous region of the Eastern Sayan, where the rivers are maximally fed due to rainfall, melting snow, and icing in summer [27].

The assessment of long-term fluctuations in the average and maximum discharge was carried out using difference integral runoff curves, which reflect long-term periods of increased and decreased water content. The data of the following hydrological stations on the left tributaries of the Angara were used: (1) Kitoi—Kitoi workers' settlement; (2) Kitoi—Dabady village; (3) Irkut—Tibelti village; (4) Belaya—Mishelevka village; (5) Irkut—Tunka village; (6) Iya—Tulun town (Figure 1). We considered the series of maximum annual water discharges for the period of observations at gauging stations. The information basis of the work is the data of Roshydromet, including from the following sources: the Automated Information System for State Monitoring of Water Bodies (AIS SMWB) [28], R-ArcticNET [29], reference materials of the State Water Cadaster, and hydrological yearbooks.

Analysis of the dynamics of planned channel deformations between 1977 and 2021 was carried out using Landsat satellite images (MSS, 5 TM, ETM+, OLI, and TIRS) and topographic maps. The most suitable combination of channels for Landsat 5 is 7,4,2, and 7,5,3 in Landsat 8. This combination gives an image that is close to natural colors and allows you to identify water bodies.

For the reliability of the analysis of channel deformations, we used images with the dates of shooting in a clearly-defined summer–autumn low-water phase, excluding periods

of summer floods. Correlation, digitization, and analysis of multi-temporal cartographic layers were carried out using GIS MapInfo.

In the geomorphological regions under consideration (mountains, piedmonts, and plains), the areas of the floodplain islands within the channel, as well as floodplain–channel branchings for different observation periods, were calculated. Further, the areas within the boundaries of the districts were reduced to a dimensionless form, where "1"wastaken to be the total area of the island floodplain in 1977.

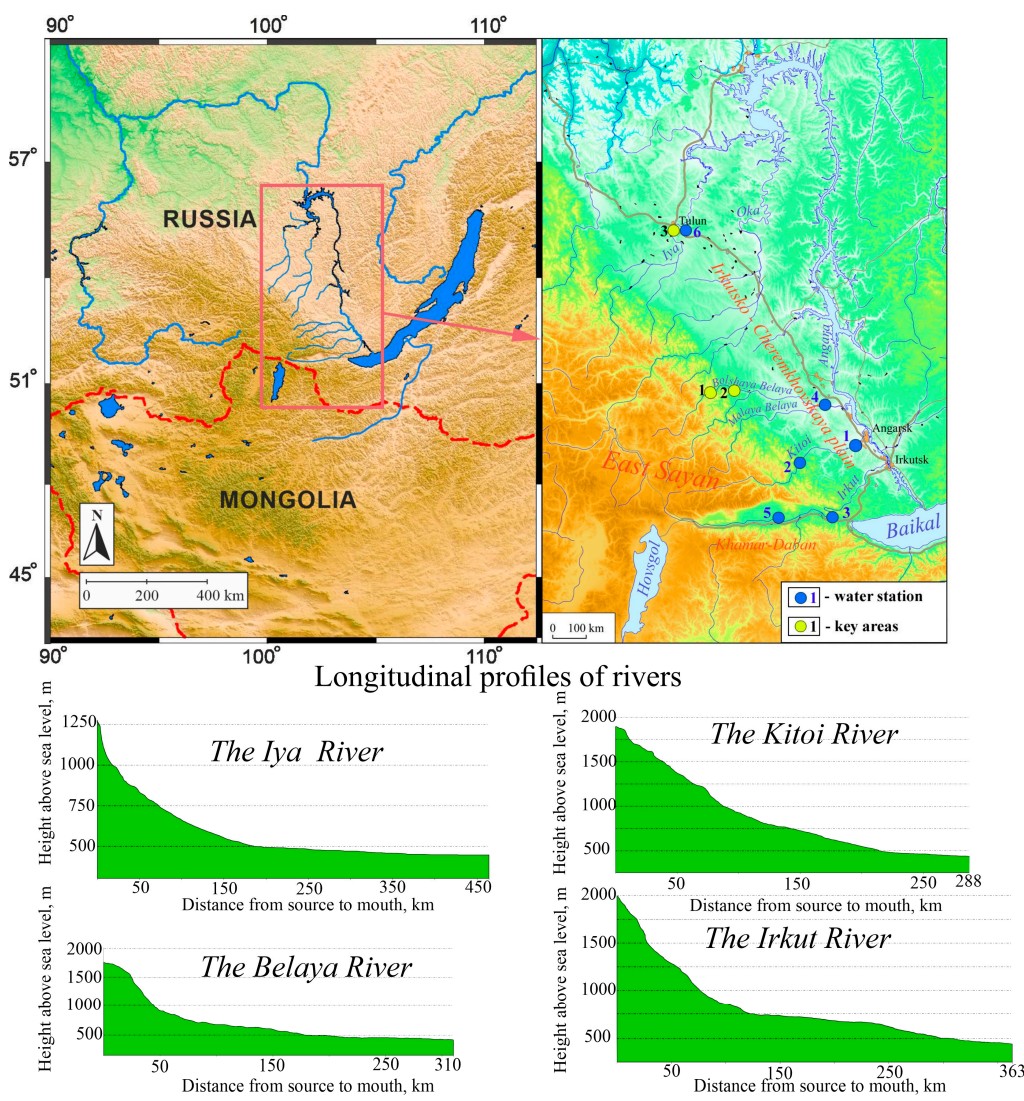

**Figure 1.** Location of studied rivers and water stations: 1—Kitoi–Kitoi; 2—Kitoi–Dabady; 3—Irkut–Tibel'ti; 4—Belaya–Mishelevka; 5—Irkut–Tunka; 6—Iya–Tulun. Key areas: 1—Novostroika, 2—Bolshebel'sk, 3—Tulun.

The typification of floodplain–channel complexes of rivers was carried out based on the ideas of the Russian school of geographical channel studies [2,11,30,31]. Multi-year research, including monitoring observations of channel deformations and standard observations, yielded factual evidence of morphological and morphometric parameters and coast dynamics in low and high-water periods [22].

To assess the geomorphological situation and obtain up-to-date information during floods, quadcopters (DJI PHANTOM4 Pro, Mavik) were used, which made it possible to qualitatively record the current situation in the river valleys and environmental changes under the influence of natural and technogenic factors with a high degree of reliability.

Tostudy of the physical and mechanical properties of soils, granulometric analysis was carried out at the laboratory of the Institute of the Earth's Crust SB RAS.

## 3. Results and Discussion

### 3.1. Maximum Discharge and Floods: Generation Conditions and Spatiotemporal Dynamics

The main natural factors in the development of the channel and the formation of the floodplain are the hydrological regime of the river and the flow of the channel-forming turbidity current [2,10,11]. The most intensive channel processes occur in the water-abundant phase of discharge, when the rivers are at their maximum water discharge during spring floods and summer rain floods. At this time, floods occur tothe rivers. A significant part of the flood-driven damage is directly related, not to the fact of flooding, but to intense channel deformations and flood-forming processes.

The regime of the rivers on the left bank of the Angara is mixed: in the upper reaches, their flow is formed within the Eastern Sayan and is characterized by the predominance of rain floods in the warm period of the year. Floods, as a rule, follow one after another; the water content of rivers is high during this period. Usually there are from one to four floods during the warm season of the year, more often two or three. Spring high water in the upper reaches is not very pronounced. As we move downstream, the role of rain floods decreases and, at the same time, the role of spring floods increases. Spring high water is very noticeable in the lower reaches of the rivers, differing from rain floods by a significant decrease in levels and flow within 20–30 days. On the Irkut River, the spring high water is not pronounced almost throughout the entire length of the river.

The territory under consideration is characterized by a high risk of flooding [32]. The greatest danger exists for the most populated and economically-developed piedmonts in the middle and lower reaches of the left-bank tributaries of the Angara River, originating in the Eastern Sayan mountains. The main causes of floods here are rain and mixed floods (rain and snowmelt in the mountains). They occur here most often in July, less often in June and August. The largest number of flooded settlements in the Angara basin are located on its left tributaries in Nizhneudinskii, Taishetskii, Usol'skii, Tulunskii, Shelekhovskii, and Ziminskii districts in the basins of the Biryusa, Uda, Oka, Iya, Belaya, Kitoi, and IrkutRivers. In these basins, 137 settlements are at risk of flooding, including the towns of Tulun, Nizhneudinsk, Irkutsk, Zima, Angarsk, and Shelekhov.

These areas have long suffered from floods. Historical materials [23,33] recorded more than 70 cases of rain floods in the left tributaries of the Angara. Most often, they occurred in June and July. The flood of the Iya River was first mentioned in 1820. Then, it occurred in the Lena, Angara, Irkut, Iya, and Dzhida over an area of more than 150,000 km$^2$ [23]. A similar flood occurred in 1870: "... on 23 June 1870, a flood in Iya and Azeya caused damage to the peasant farms of the town of Tulun. On June 25, the Biryusa flooded. On July 20–22, it rained for two days, water rose in the Irkut and Angara rivers, flooding the bridge on the Irkut and the meadow to the Ascension Monastery. In July, floods occurred on Uda, Iya, Azeya, Oka, Biryusa, and Belaya over an area of more than 100 thousand km$^2$" [23] (p. 279). Floods of this magnitude have occurred more than once in the left tributaries of the Angara in the recent past. Over the past 50 years, the following major floods have brought the greatest damage to the economy and population:

- a catastrophicflood in the summer of 1971, when 33 settlements, 82 industrial enterprises, and about 700 km of roads were flooded only by the rivers of Irkutsk Oblast;
- a flood in July 1984 in the Tulun district. In the IyaRiver basin, 12 settlements were flooded (including one-third of the town area), 5.5 thousand hectares of pastures and approximately 800 hectares of crops;
- a flood in 7–12 July 2001. Many areas of Irkutsk Oblast suffered from the flood in the rivers Kitoi, Irkut, Belaya, Iya, Oka, and their tributaries, most severely in the Ziminskii district. More than 150 settlements with a total population of 460 thousand people were flooded. A total of 7towns were flooded, 11 people died, and 12 thousand

people were evacuated. The damage was estimated at 1.75 billion rubles. The cause of the disaster was heavy rain, which exceeded the monthly norm for several days;

- in June and July 2019, catastrophic rain floods occurred in the Angara's left-bank tributaries, the rivers Iya, Uda, Oka, and Biryusa. Eight districts of Irkutsk Oblast were flooded. A total of 109 settlements suffered from the flood, as did hundreds of kilometers of roads and crops, 22 bridges were demolished or damaged, 26 people were killed, and 5 were still missing. The biggest impact was on the town of Tulun, which sank into the water to a depth of several meters. At the end of July 2019, the second flood passed. The floods of 2019 were caused by extreme precipitation on the pre-moistened surface of watersheds [34].

Long-term fluctuations of the maximum annual runoff were assessed using differential integral curves that reflect long-term periods of increased and decreased water content (Figure 2).

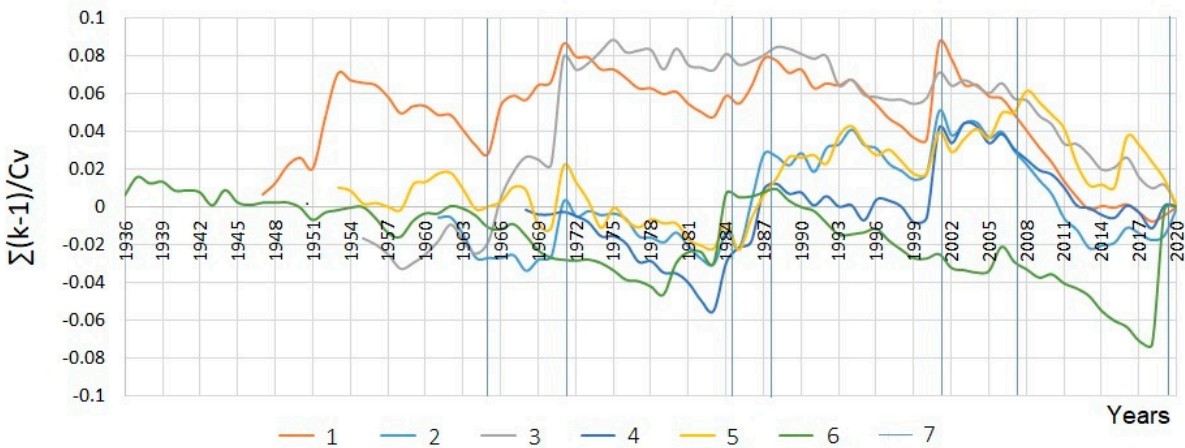

**Figure 2.** Long-term changes in the maximum annual discharges in the Angara River basin in the form of differential integral curves; k—modulus code, Cv is variation coefficient. 1—Kitoi–Kitoi, 2—Kitoi–Dabady, 3—Irkut–Tibelti, 4—Belaya–Mishelevka, 5—Irkut–Tunka, 6—Iya–Tulun, 7—periods of water content.

The analysis showed that, despite the existing differences in long-term runoff fluctuations of the rivers of the Angara basin, there is some synchronism in their fluctuations. As a result, there was a high-water flow phase in the rivers from the early to mid-1960s until 1971.The most abundant year was 1971, when floods occurred in several rivers (most notably the IrkutRiver), and then a gradual decrease in maximum annual discharges was observed in the rivers until 1983. Between 1983 and 1988, there was an increase in water content, except for the IrkutRiver, where the average water content was maintained from 1971 to 1992. The period between 1989 and 2001 wascharacterized by relatively high maximum discharge values ofall rivers, with a slight decrease after 1994. In 2001, a catastrophic flood occurred in the left tributaries of the Angara, after which the period between 2001 and 2006 was characterized by increased water content. After 2007, a low-water period began, and the maximum annual discharges decreased until the high-water year 2019, when an extreme flood occurred in the Irkut, Kitoi, Belaya, and IyaRivers. As a result, six periods of water content change were identified according to the maximum annual discharges for the period spanning from the start of observations until 2019 forthe rivers under consideration.

For each series of maximum annual discharge, empirical supply curves were constructed, variants of its analytical approximation were selected using the methods recommended by SP 33-101-2003 [35,36], and the calculated values of the maximum annual discharge of standard supplies were obtained. The graphs show the values of the quantiles of the probability curve of the maximum annual runoff: 1%, 2%, 5%, 10%, 20%, and 50%

according to the Pearson curve (type III), with the estimation of parameters using the method of moments (Figure 3).

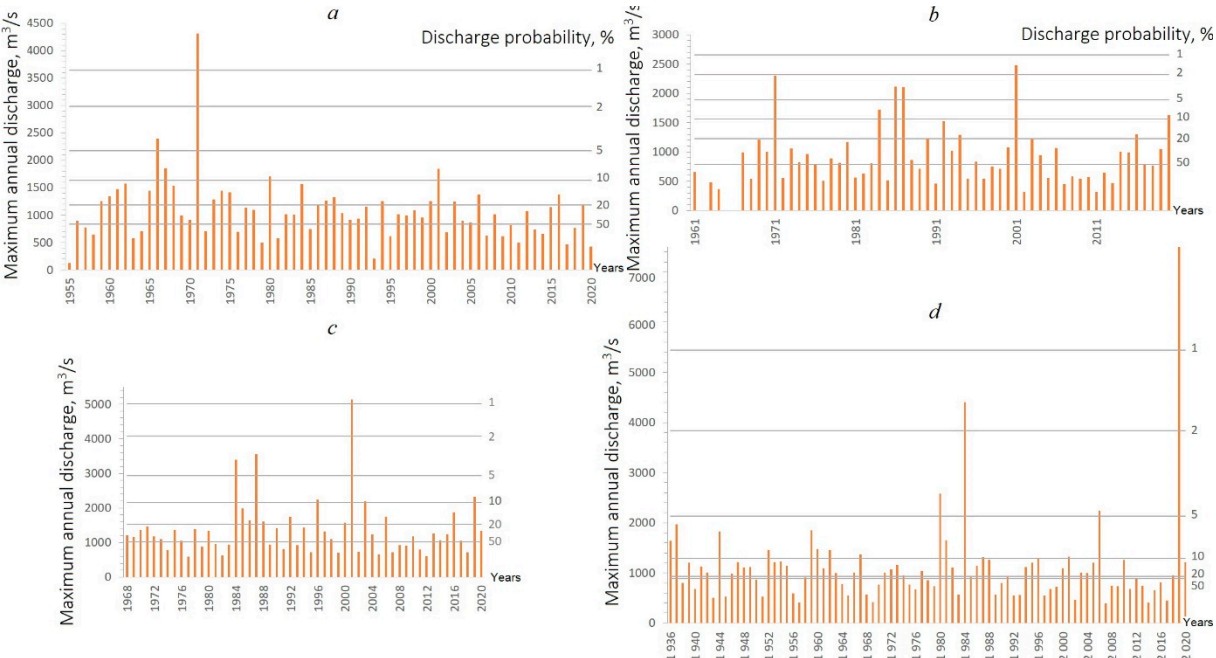

**Figure 3.** The observed maximum annual water discharges and the values of their standard probabilities calculated according to thePearson curve (type III) with parameter assessment using the method of moments: (**a**)—Irkut–Tibelti, (**b**)—Kitoi–Dabady, (**c**)—Belaya–Mishelevka, (**d**)—Iya–Tulun.

Floods of these rivers were caused by heavy rains in 1971, 2001, and 2019. Thus, the probability of floods in 1971 was about 0.5% at the IrkutRiver gauging station (the village of Tibelti) and about 2% at the KitoiRiver gauging station (the village of Dabady). The probability of floods in 2001 was about 1% for the Belaya River (the village of Mishelevka) and about 1.5% for the KitoiRiver (the village of Dabady). During reconstructions, using different methods, of the catastrophic flood of 2019, the value of the maximum discharge at the gauging station of the IyaRiver (the town of Tulun) was estimated by various experts to be in the range of 5640–6100 m$^3$/s. [37–39]. At the same time, Roshydromet set the value of the maximum flow rate much higher, 500 m$^3$/s, and we have to operate with it. [28]. The reliability of this flow rate, depending on the approximation method, varies from 0.1, according to the Kritsky and Menkel distribution with the assessment of parameters according to the maximum likelihood estimation, to 0.8, according to the Vinogradov C-3 curve in terms of accuracy. The reliability of this flow rate is about 0.5% according to the Pearson type III distribution with the estimation of parameters using the method of moments.

### 3.2. Distribution of Morphodynamic Types of Channels and Manifestations of Channel Deformations

The dynamics of floodplain–channel complexes amid increased water content were analyzed for the Irkut, Kitoi, and Belaya Rivers (Table 2). To determine the functioning of floodplain–channel complexes in various hydrological phases, a typification of morphodynamic channel types was initially carried out. Next, the areas of floodplain–channel complexes directly involved in the flow–land interaction werecalculated, and the mechanisms and factors for the development of the most dynamic areas were determined. The floodplain area in 1977 was taken as the reference point, which wascharacterized as the medium water one. Further, the areas in 1995 (low water) and 2021 (relatively high water) were estimated.

**Table 2.** Distribution of morphodynamic types of channels in the area under study.

| Name of the River | Belaya | | Kitoi | | Irkut | |
|---|---|---|---|---|---|---|
| Channel Type | Length, km | Length, % | Length, km | Length, % | Length, km | Length, % |
| Wide floodplain | 212.60 | 59 | 185.75 | 59 | 285 | 58 |
| Incised | 45.60 | 13 | 59.00 | 19 | 145 | 30 |
| Adapted | 100.80 | 28 | 71.25 | 22 | 58 | 12 |
| Overall Length | 359 | | 316 | | 488 | |

The main morphodynamic channels were typified according to three main types—wide floodplain, incised, and adapted channel types—where the floodplain width is 2–3 times the channel width [30].

To identify the dynamics of planned channel deformations within the identified geomorphological regions, we calculated the areas of the floodplain islands for different periods of observation (Figure 4).

The total area of insular land mass in 1977 for each site (mountain, piedmont, and plain) was taken as a unit. Variations occurred around the figure of one (Sr).

The Belaya River best illustrated the dynamics of the channel–floodplain system (Figure 4A). For the Belaya River, the minimum values of the long-term variability of the floodplain area were revealed. For the mountainous part, where incised and adapted branched channel types were predominantly developed, a decrease in the floodplain area was observed due to the flooding of the islands from 1995 (the low-water period) to 2021 (the high-water period).

The increase in the area of the islands in the piedmont area was associated with the death of the transverse channels in the dry period and the shallowing of the banks. In 2021, the area of the floodplain, on the other hand, increased due to the revival of shoals. In the lower stretch with a dominant adapted channel type, the area of the floodplain islands increased in 1995 due to shallowing. In 2021, the area of the floodplain decreased due to the fragmentation of insular landmasses by minor channels and their flooding.

A significant increase (almost twice as large) in the floodplain area occurred in the mountainous part of the KitoiRiver (Figure 4C) within the channel sections with significant slopes of 5–12‰. Downstream, in the braided reach, the area changes wereinsignificant, increasing with the growth of islands in a relatively dry period and decreasing due to the reduction in the area of islands during flooding in a high-water period. The semi-mountains and floodplains with anabranching channels werecharacterized by the increasing area of the floodplain islands.

The IrkutRiver's mountainous section was distinguished by the development of adapted and incised channel types within the mountainous section, as well as wide floodplains and adapted within mountain–hollow areas. Here, there is an increase in the floodplain area during high-water periods due to the activation of distributary channels (Figure 4C). The floodplain islands within the piedmonts werepractically stable. This wasexplained by the distribution of an incised, predominantly non-braided channel type here.

Within the plain–platform part of the current, the development of a wide floodplain channel type with forced meanders, a floodplain, and single simple and complex types of anabranches was characteristic. This type of channel occupied 60.4 km (93.3%), and the areas with a braiding of river channels accounted for 15 km (24%), while the rest of the reach had an adapted channel type [40].

The main types of horizontal deformations of the channels under study were as follows. These werespatial manifestations of areal (reformation of meanders) and local (changes within meander elements) death of transverse channels at floodplain forks, increase and decrease in island area, and erosion and aggravation of banks [40,41].

Thus, it is possible to note the commonalities of channel processes, which were revealed in the analysis of channel deformations of rivers located in different geodynamic

settings. The largest increase in the area of floodplain islands was characteristic of gravel-dominated and laterally-anabranching channels. There is an increase in the area of flood-plains from the mid-water period to the high-water period in the piedmont of the Kitoi and Belaya Rivers. The piedmont is an area of internal deltas with the development of an anabranched and braided type for these rivers.

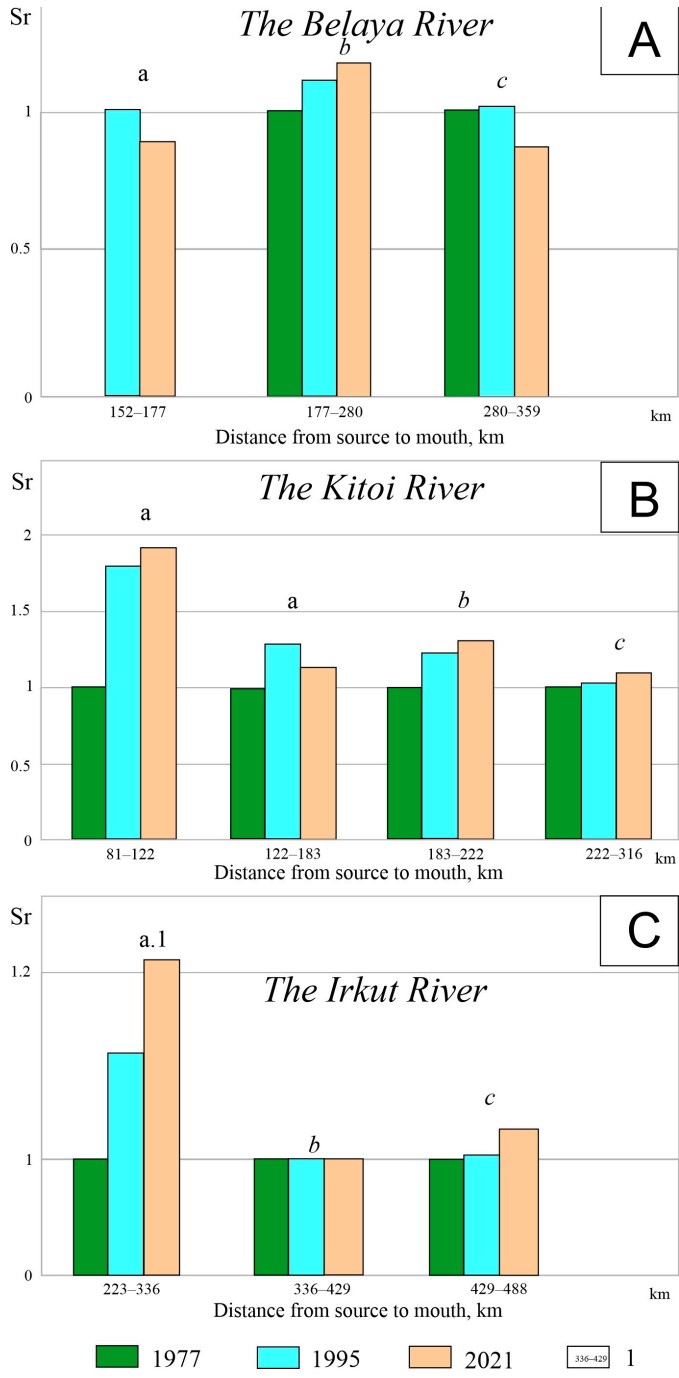

**Figure 4.** Diagram of changes in the area of the floodplain islands between 1977 and 2021. Dynamics of the area of the floodplain islands of the left-bank tributaries of the Angara River between 1977 and 2021 in various geodynamic areas (periods: 1977—medium water; 1995—relatively high water and 2021—high water (**A**)—The Belaya river; (**B**)—The Kitoy river; (**C**)—The Irkut river). Areas (Sr) are given in relative units (the floodplain area in 1977 is taken as 1); 1–distance from the riverhead.

The values of the relative variability (ΔSr) of the floodplain islands within different geodynamic areas for 1977–1995 and 1995–2021 (Figure 5) showed that the mountains and piedmonts werewithin the same interval of0.1–0.4. The exceptions were the minimum and maximum values of0 and 0.7, respectively, near the IrkutRiver in the piedmonts and the KitoiRiver in the mountains.

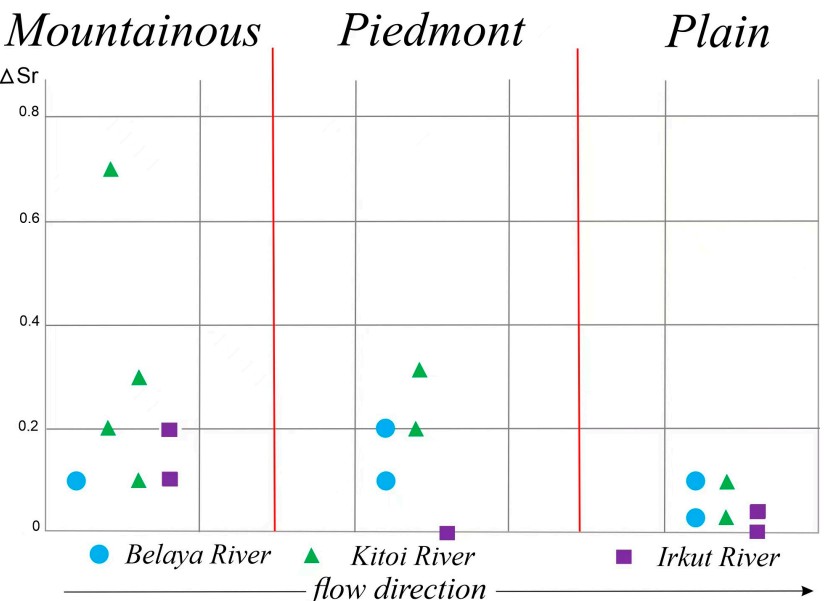

**Figure 5.** Distribution of indicators of relative variability of floodplain islands of the left-bank tributaries of the Angara River from 1977 to 2021 in various geodynamic areas.

The piedmont of the Irkut basin correspondedmorphostructurally to the shoulder of the Baikal rift. The incised channel here was stable, but, with an increase in the water level, individual channel bars flooded. The maximum variability index of the Kitoi floodplain area was probably due to the morphometric parameters of the relief, significant dissection, and steepness of slopes (caused by the high intensity of slope processes), as well as the confluence of the large Shumak tributary and the location of the river section between two fault zones that limit the structural stage.

The smallest fluctuations in the areas of floodplains (interval rate of 0 to 0.1) in the plain part were typical for all the rivers considered. For rivers with a wide floodplain type, the development of adapted meanders and reaches with two-arm channels (Irkut and Kitoi) led to a general increase in the area of island floodplains. The Belaya River within the plain was characterized by the development of adapted and incised channels, as a result of which the floodplain islands have been reduced.

*3.3. Impacts of Floods and Activation of Dangerous Geomorphological Processes*

The complexity of the morphotectonic structure, discontinuities of various orders, and heterogeneity of the lithological composition of rocks within the Irkutsk–Cheremkhovo Plain determined the diversity of morphodynamic types of river valley channels. The influence of these factors most clearly manifested itself in a combination of wide floodplain, adapted, and incised channel types within the valleys of the left bank tributaries of the Angara River [22].

3.3.1. Plains

The nature of freshets and inundations resulted from the development of the morphodynamic type of the channel and floodplain, as well as their combination in different sections of the flow, is illustrated by the examples below.

The IyaRiverwas characterized by such an alternation: a wide floodplain channel when crossing sandstone fields and an incised or adapted type when the river crossed Triassic traps. During the second stage of the summer flood of 2019, three sites in the IyaRiver valley were studied. The first site was located near the village of Gadalei. A wide floodplain-type channel with free meandering has developed here. During the second flood stage, the negative forms of the floodplain microrelief were flooded (Figure 6a).

*a*

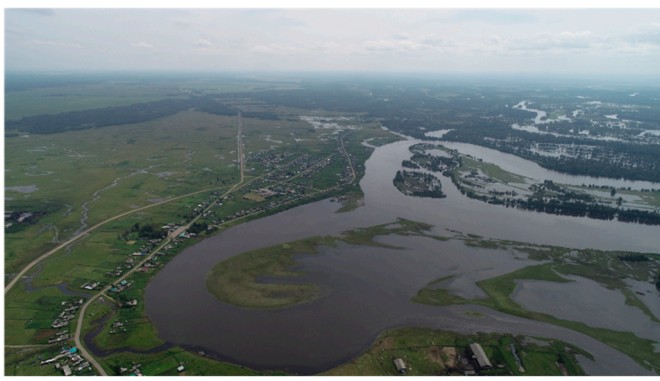

*b*

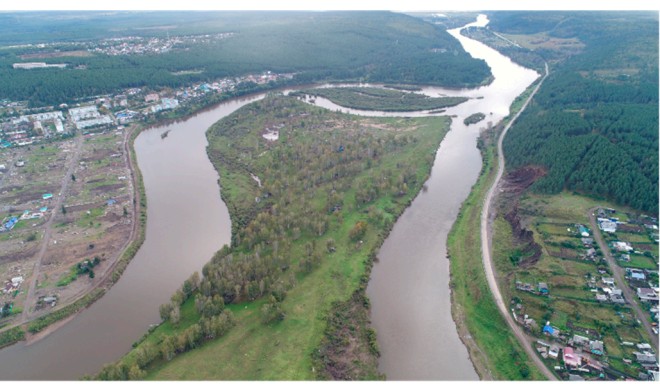

*c*

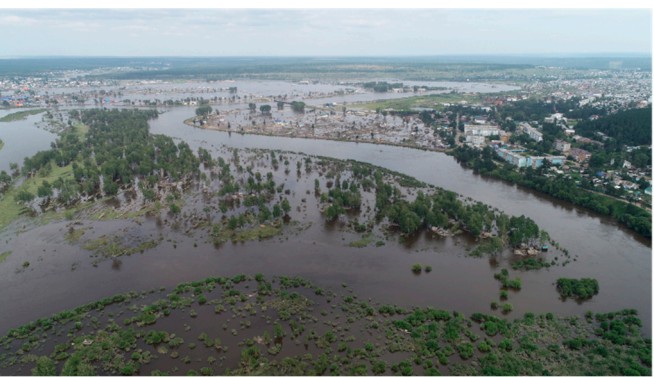

**Figure 6.** Impacts of floods on the IyaRiver: (**a**)—flooding of the wide floodplain section of the valley in the vicinity of Tulun; (**b**)—consequences of bank erosion; (**c**)—processes of bank destruction and landslides (area of the gauging station in Tulun). Survey coordinates: 54°24′27.4218″ N, 100°43′23.1820″ E. Shooting: 2 August 2019.

The second site, with the development of an adapted-type channel, was studied to the south of the Hydroliznyimicrodistrict in the town of Tulun, near the pumping station.

A narrow, even floodplain, the width of which was equal to 2–3 times the channel width, turned out to be completely flooded. The alternation of different types of channels and floodplains and different capacities of floodplains affects both the types of interaction between the channel and floodplain flows and the types of erosion-accumulation processes [42,43]. Thus, horticultural cooperative territories located on the adapted bends upstream from the Hydrolyznyimicrodistrict became trapped. The swelling of the tidal wave after the wide floodplain of the Gadalei expansion of the IyaRiver valley served as an additional factor that increased the strength of the stream, which literally swept away all the buildings. Smirnov and Tkachev [44] proposed to call such phenomena "areal river erosion". This type of erosion is defined as the flat washout of buildings and engineering structures by river waters during periods of torrential floods and short-term natural and artificial floods (Figure 6b). In the microdistricts of the town of Tulun, which were most affected by the flood, the activation of erosion processes in the bank line was recorded, in particular the formation of subsidence fractures, the processes of collapse, sliding of blocks, and linear erosion (Figure 6c).

Either way, flooding of territories causes the activation of exogenous processes, which is unfavorable for human life. This is especially true for urban areas.

On the surface of terrace II (10–15 m) composed of boulder-pebble, sand, sandy loam, and loams (8–15 m), we recorded landslide deformations that led to foundation deformations (54°34′13.93″ N 100°35′41.74″ E) (Figure 1). The southern and central parts of the site hadthe highest concentration of deformations and a series of fracture zones (Figure 7a).

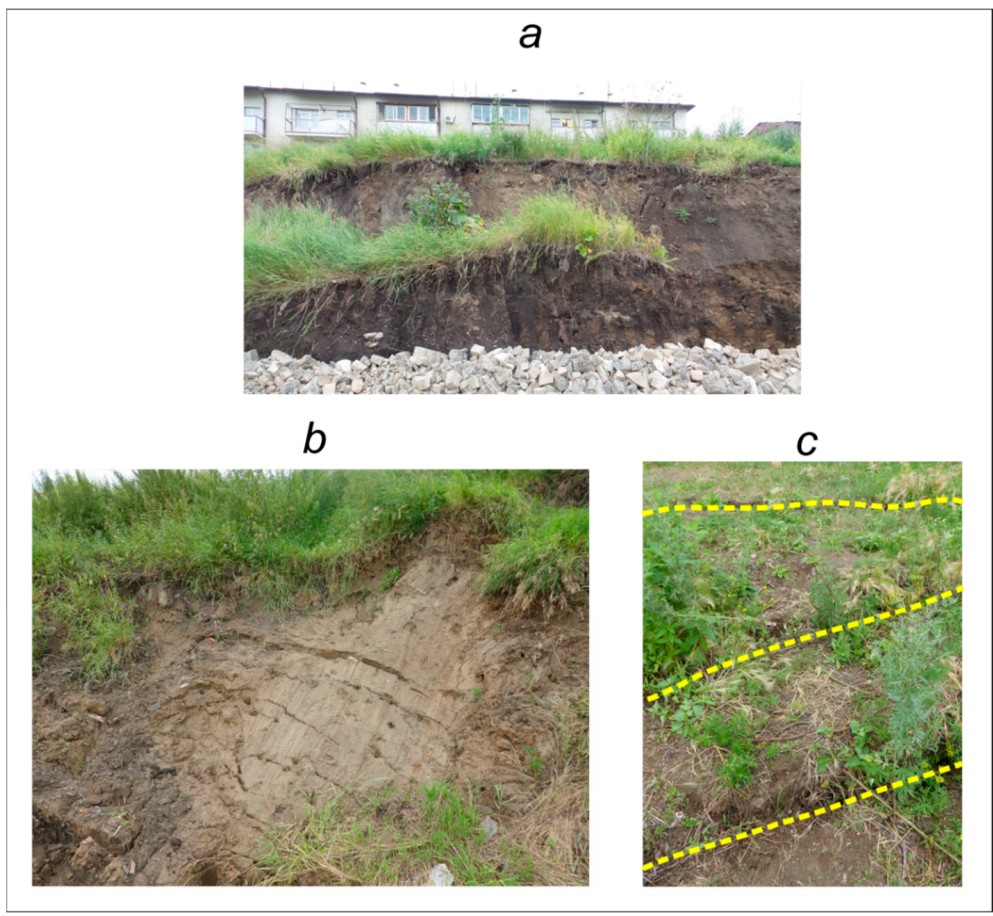

**Figure 7.** Landslide deformations of the bank line of the second terrace of the IyaRiver (the town of Tulun): (**a**)—frontal part of landslide deformations; (**b**)—landslide block main scarp; (**c**)—extension fracture on the surface of the second terrace.

On the terrace, in close proximity to a residential development, in the step zone, a series of strike-slip tension joints was noted. On the edge of the step zone, subsidence of blocks and local soil displacements were noted (Figure 7b). There was a series of blocks separated from each other by cracks (width from 1.12 to 1.8 m), and extension cracks opened up to 0.25 m (opening depth of up to 0.3 m) (Figure 7c).

The site was located at the top of the river bend and the main deformations were noted here. During the flood, the strength of the main water flow fell exactly on the site where residential buildings were constructed (Figure 8). Further, the flow, losing its strength from hitting the shore, turned almost 90° and went downstream (Figure 8). During the flood, the dead arm of the river merged with the general stream, and the main channel performed the most intensive erosive work. The analysis of satellite images showed that it was the river reach that concentrated the main stream.

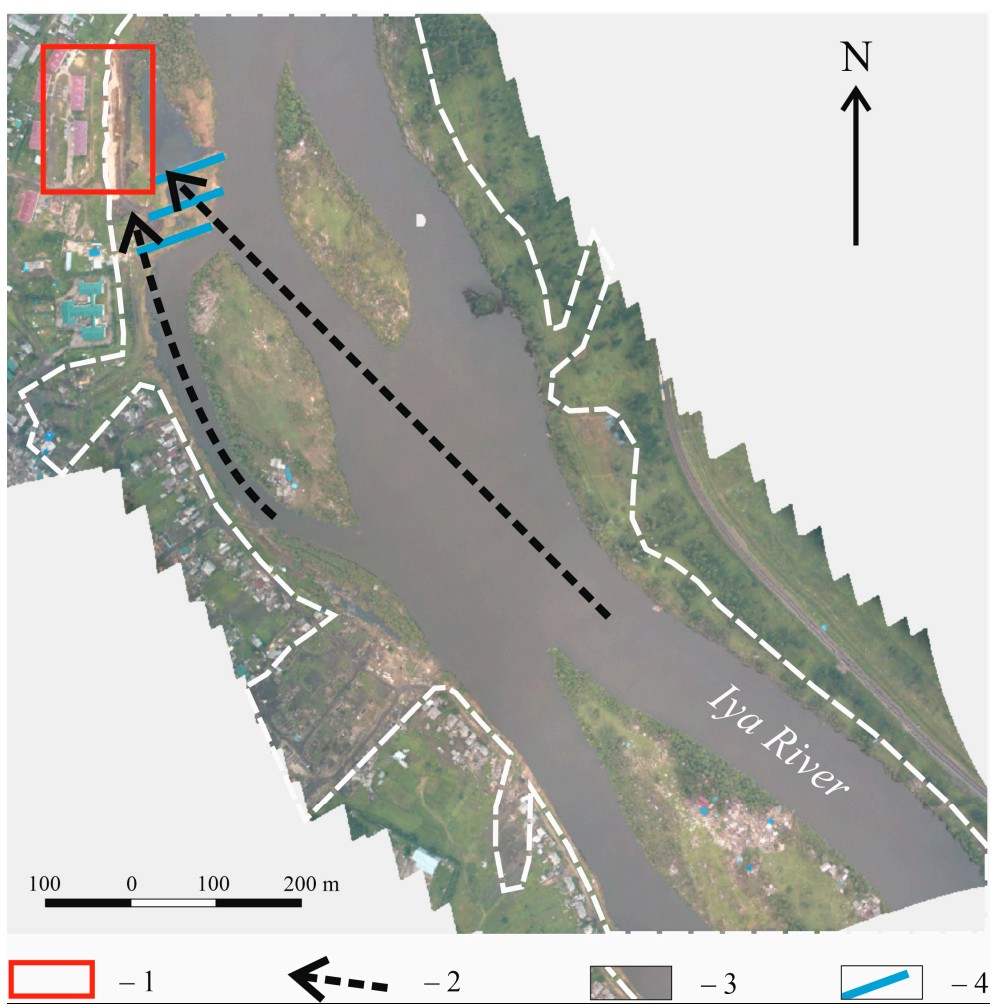

**Figure 8.** Set of the current of the IyaRiver in the area under study. 1—area under study; 2—set of the current of the IyaRiver; 3—border of the flood zone (7 July 2019); 4—bank protection structure.

Directly in front of this bank section (upstream of the river), bank protection dams had already been erected. Before the flood in July 2019, the dams performed their function properly, preventing the erosion of the bank and contributing to the accumulation of material behind the dams. However, the height of the dams was not designed for the level of this flood. During the flood in July 2019, the level of the IyaRiver exceeded the level allowed for protective structures. The elevation of the dam crest was 457 m, while the level of the IyaRiver exceeded 464 m in this reach.

According to laboratory studies of the composition, state and properties of dispersed soils, deposit zones that were the most sensitive to various deformations (compaction, liquefaction, fluidity, thixotropy, etc.) were identified. They turned out to be technogenic formations and a complex of undivided diluvial and alluvial deposits.

A common feature of technogenic deposits is their heterogeneous composition in area and section (argillaceous, sandy, and blocky material), as well as their predominantly low soil density and high porosity. Deconsolidation of soils was caused by the suffusion removal of finely dispersed material by water flow and subsequent liquefaction of soils, which led to the formation of weakened zones and the development of landslide deformations. The overlying blocks subsided and slipped towards the river as a result of the outflow of a water-saturated, weakened soil horizon.

### 3.3.2. Slope Processes and Coastal Deformations in Mountains and Piedmonts

Increased channel deformation, hazardous processes, and the intensity of their manifestation distinguish mountains and piedmonts [36]. Erosion processes in the piedmont of the areas under consideration caused the destruction and retreat of bank lines, sheet floods and linear erosion (formation of gullies and traces of erosion) on the surface of the floodplains.

Substantial rainfall, as well as rain floods, undoubtedly contributed to the intensification of exogenous geological processes. The water saturation of soils in the intensification of slope processes of various classes produced the quickest response. Slope processes weremost extensive and intensive in the mid-mountain and foothill regions of the East Sayan uplift. Steep slopes were indicative of a wide range of slope processes that have developed. Therefore, in the area located above the village of Arshan, with a length of 500 m, on slopes up to 150 m long, a steepness of 20–25°, and the relative height of 30–40 m at the river's edge, a series of debris flows was formed. The sediments here wererepresented by loose, blocky-crushed stone-land waste deposits up to 2 m thick. During rains, water flows, concentrating in the runoff troughs, saturated the weathered deposits and turned into a fluid state, and the water-saturated mass slid into the riverbed (Figure 9a). In the upper part of the slope, a debris flow formed. In the middle part, the processes of collapse and shedding joined it. In the lower part of the slope, which was in interaction with the river flow, the formation of a skeletal floodplain began—a step niche with a thin layer of alluvial deposits (sand and pebbles) formed here. In addition to these newly-developed forms, an erosion furrow could be noted as an indicator of a mudflow.

Sod layer creeping processes developed in the lower parts of the slopes (Figure 9b), scarps of floodplains, and low terraces (Figure 9c), which later often served as a kind of brow and prevented the scarp from being destroyed by the river flow.

The processes of urbanization in the mountains and piedmonts of the territory wereinsignificant; however, the influence of infrastructure facilities often provoked an extreme manifestation of channel processes.

The most intense manifestation of erosion-accumulation processes was observed in the areas where the channels werecrossed by dirt roads, which played the role of dams during the watering of the territory. The Bolshaya Belaya River valley was characterized by broad floodplain–branched–sinuous channels with adapted and free bends in the piedmonts. Within the piedmonts, the Belaya River wasa gravel-dominated, laterally-anabranching river.

Roads within the area under consideration often ran along the channel, crossing numerous anabranches. Because these channels weretypically drained during low-water phases, bridge crossings werelimited to road filling. In two areas, Novostroika and Bolshebel'sk (Figure 10), on the surfaces of a high floodplain, we observed a reshaping of the relief caused by extreme processes of relief formation associated with the destruction of roads by floodplain streams. A high floodplain of 2–2.5 m in the near-river part wascomposed of a pebble bed 1.3–1.5 m thick, with sandy loam deposits up to 0.7 m thick on it.

*a*

*b*

*c*

**Figure 9.** Slope processes of the IyaRiver valley in the piedmont of the Eastern Sayan: (**a**)—debris flows; (**b**)—debris slides in the lower part of the root slope; (**c**)—sliding of the soil in the scarp of the floodplain.

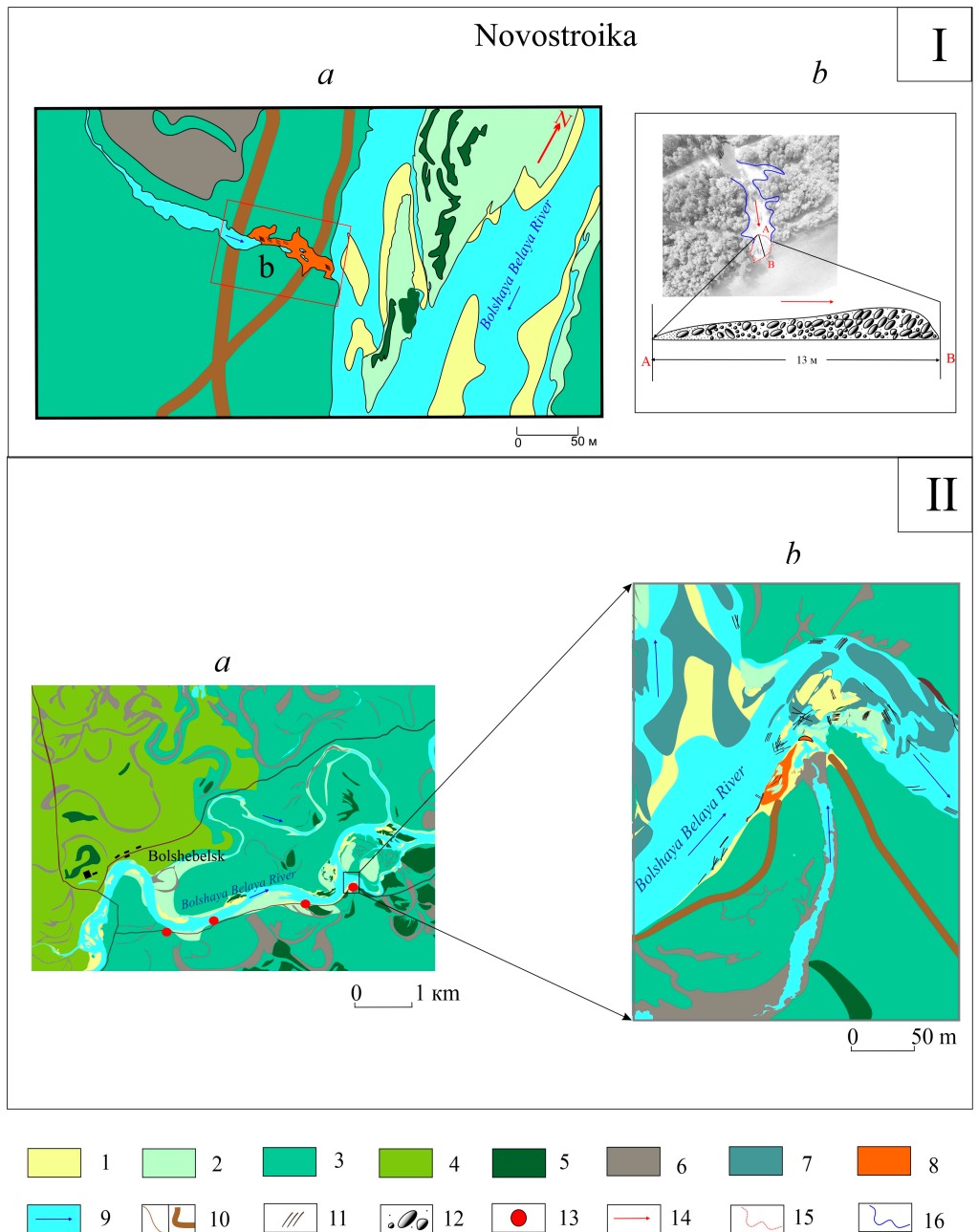

**Figure 10.** Floodplain–channel complexes of the Bolshaya Belaya River in the area of observation of bank deformations. (**I**) Novostroika: a—an index map compiled according to the orthophotomap (shooting date July 2019); b—an aerial photograph (shooting date July 2019); line AB—a longitudinal profile of the channel-mouth bars. (**II**) "Bolshebel'sk": a—an index map compiled from a satellite image of Google Earth (shooting date June 2019); b—an index map of the relief structure of the site, compiled according to the orthophotomap (shooting date July 2019). Legend: 1—sand and pebble shallows; 2–low floodplains up to 1 m high; 3—high floodplains 2.5 m high; 4—the first terrace up to 6 m high; 5—floodplain crests up to 1.5 m high; 6—hollows eroding up to 1.5 m deep; 7—channel ridges; 8 –a section of relief reformation and formation of the channel-mouth bar;9—water bodies, set of current, section of the road destruction, and the formation of the channel-mouth bar; 10—roads; 11—areas of wood accumulation; 12—sandy-pebble deposits; 13—monitoring points for coastal processes; 14—current directions in case of a road break; 15—outlines of the channel-mouth bar; 16—boundaries of flow distribution.

The flow at the Novostroika site was saturated with coarse clastic material and its further redeposition at the mouth was a result of the road's backwater effect, further breakthrough of the embankment body, and destruction of the road (Figure 10Ia).

In the channel's mouth, an accumulative body formed of swell-shaped pebbles with a longer slope towards the river (steepness from 15° to 24°) and a steep bank scarp (30–31°) (Figure 10Ib) formed in the mouth part of the channel. The channel-mouth bar wasa ground fragment of the alluvial fan, which extended into the channel from 2 to 4 m, with an area of 27 m². The dimensions of its above-water part were 13×5×1.5 m; that is, the volume of the alluvial cone was approximately 130 m³. The area of its underwater part, according to the analysis of the orthophoto, was about 162 m², but it was difficult to estimate the initial volumes of the removed material. In the mouth part of the body of the bar, small channels werecut through, 35–40 cm deep and up to 1.5 m wide.

The strength and speed of the breakthrough flow could be judged by the structure of the channel gravel and the morphology of the channel below the destroyed bridge. The length of the channel was 40 m, with a width of 4–5 m. In the channel, water-breaking niches up to 1 m deep and 10–11 m long were noted. The bars formed along the periphery of the channel and were composed of pebble material up to 1.2 m high and up to 13 m long. The deposits formed during the distribution of the flow below the destroyed road embankment were 20 cm.

The pebbles in the area below the break of the dam-bridge were laid vertically (the long axis was oriented perpendicular to the surface), which is typical for mountain rivers with high flow turbulence. The calculated flow velocities for such values of the average pebble diameter (6 cm) and the average flow depth of 1.5 m could exceed 2.4 m/s [45]. The Bolshebel'sk site was located 53 km downstream, just as a road crossed the floodplain in the preceding section (Figure 10IIa). Based on previous year's images, it could be assumed that the road construction disrupted the water exchange system of the channel and the main channel, resulting in the channel's death against the backdrop of a dry phase (Figure 10IIb).

The formation of the mouth bar in this area occurred according to the scenario described above: the channel was filled with water, the road embankment broke, and the alluvial fan formed. The area of the ground part of the alluvial fan, the mouth bar in this area, was 185.75 m², the width reached 10 m, the length was 28 m, and the height from the edge was up to 1.5 m. Several channels up to 1 m deep also formed in the bar body, as well as a terrace up to 0.9 m high on the side of the main river.

In the summer of 2020, during monitoring studies of the territory of the re-examination of the Bolshebel'sk site, it was found that during the restoration of the road, the mouth bar was reclaimed. However, the leveled area still prevents natural water exchange between the main riverbed and the channel. In the mouth part of the channel, new channels were formed with signs of intense incision.

Upstream, there was a section with signs of intensive destruction of bank scarp (Figure 10b). The volume of material supplied to the channel (Bolshebel'sk section) was 1800 m³ on a segment of the intensively-destroyed bank scarp of a high floodplain 600 mlong with a scarp height of 2 m and an average retreat of the scarp edge of 1.5 m. The average volume of sediments of the destroyed bank that had fallen into the channel was about 300 m³ per 100 m of the bank line in natural conditions, and in the section of the road destruction it increased to 480 m³ per 100 m.

The destruction of floodplain roads occurs as a result of ponding, an increase in flow velocity during a breakthrough of up to 2.4 m/s, and an increase in the volume of transported material—2 times (section Novostroika) and 1.6 times (section Bolshebel'sk). The formation of the bars at the mouths of the channel during the period of floods wasassociated with the backwater of the main river; this factor also subsequently madeit difficult to remove material from the floodplains.

We observed significant rates of coastal retreat and areas of extreme manifestations of fluvial relief formation within the same landmasses. They werelargely associated with the disturbance of the natural surface by engineering structures.

The roads crossing the channels played the role of dams during floods. The destruction of such artificial structures during floods often increased the intensity of the manifestation of fluvial erosion-accumulation processes, making them dangerous for humans and leading to material damage. Newly-formed accumulative forms of microrelief at the mouths of channels werecreated by interaction under the influence of anthropogenic and natural factors. We can assume the repetition of such scenarios of extreme manifestations of exogenous geomorphological processes in the future: with the restoration of roads according to the previous plan and sufficient watering of the territory. The mouth bar, which obstructed water exchange between the main river's channels and the floodplain, was another factor in flow distribution and material deposition on the floodplain's surface.

### 3.3.3. Zoning of Floodplain–Channel Complexes of Rivers According to Monitoringand Urgent Observations during Floods

For the purpose of preliminary classification of floodplain–channel complexes according to the degree of development dynamics, zoning was carried out, which took into account the following factors in their formation and functioning: the magnitude of the slope, a combination of types of floodplain–channel complexes, and the rate of planned channel deformations [22,37–39].

Within the plains, the maximum rates of bank destruction were 1.5–2 m per year. Such deformations are typical for bank scarps composed of sandy loam deposits, in which the process of sedimentation of soil blocks develops with their further collapse and shedding. The most extended areas of bank destruction werelocated within the convex banks at the tops of free bends. For rivers in the mountains and piedmonts of the left bank of the Angara, it was possible to obtain the rate of bank retreat during the catastrophic floods of 2019. For mountainous areas, they amounted to 2 m in sandy scarps, and in pebbly ones they reached 6 m due to landslide processes. A zone of anthropogenic influence and significant change or relief transformation was identified directly within the plain territories. It is a strip of estuarine sections of the left-bank tributaries of the Angara River (Irkut, Kitoi, and to a lesser extent, the Belaya River).

Such an analysis of the dynamics of the development of floodplain–channel complexes in the Upper Angara region makes it possible to identify zones of dynamically-developing floodplain–channel complexes, including mountains and piedmonts. Relatively stable floodplain–channel complexes have developed mainly in the plain–platform and plain–hollow areas. Anthropogenically-disturbed territories are specially highlighted (Figure 11).

Despite the long-term development of the Upper Angara region, it should be noted that the floodplain–channel complexes of large tributaries did not undergo radical changes; there was no change in their types; for example, no change in the channel type was recorded in these rivers.

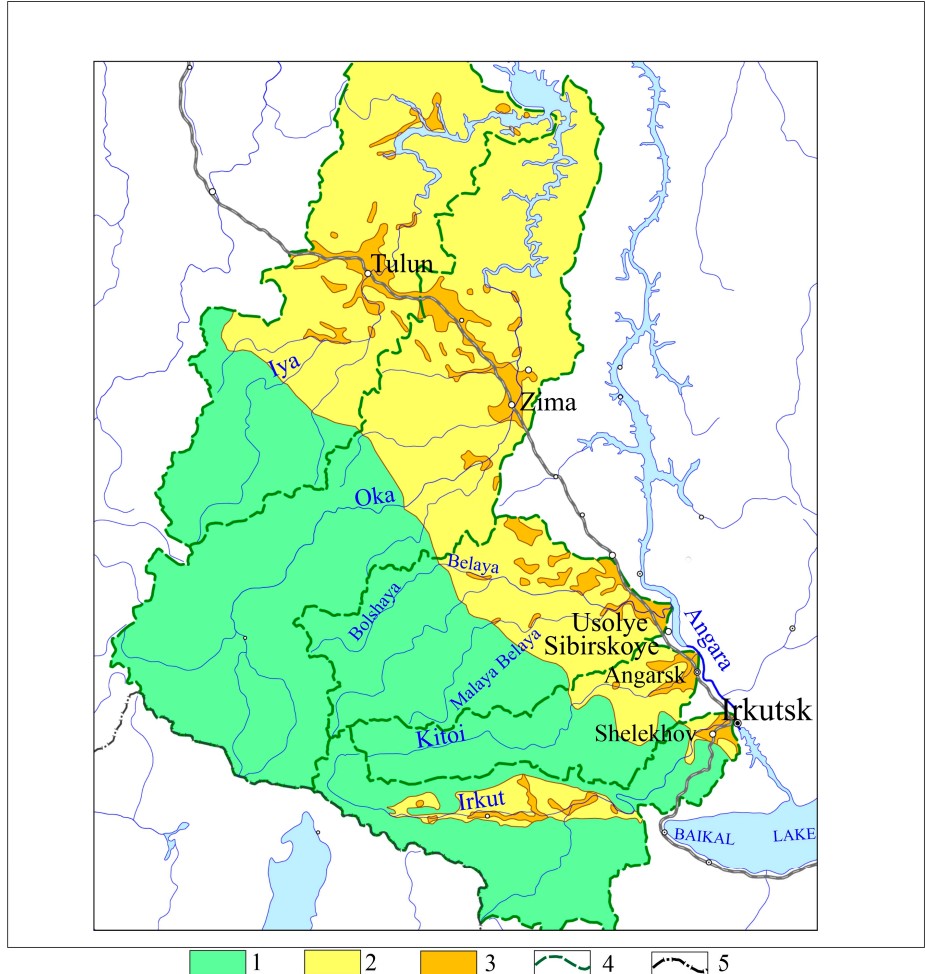

**Figure 11.** Zoning of floodplain–channel complexes along the rivers of the Upper Angara region and adjacent territories according to the degree of development dynamics. Legend: 1—dynamically-developing floodplain–river complexes; 2—relatively stable floodplain–river complexes; 3—relatively dynamically-developing floodplain–river complexes of anthropogenically-disturbed territories; 4—state borders; 5—basin boundaries.

## 4. Conclusions

The following inferences concerning the distribution of floods and hazardous exogenous geological phenomena in the research area can be made based on the information presented above:

For the considered basins, a certain synchronism of runoff fluctuations is noted. Six periods of change in water content in the rivers under consideration have been identified. The most extreme large-scale floods of the considered rivers occurred in 1971, 2001, and 2019. The findings of field observations made in 2019, immediately before, during, and after the passage of floods, demonstrate that the catastrophic floods are related to the maximum planned channel deformations and the activation of exogenous processes, especially slope activities. The occurrence of floods in 1971 (the mid-water period) varied from 0.5% in the mountain–hollow (Irkut-Tibelti) to 2% in the mountain–piedmont (Kitoi–Dabady) parts of the basins. The flood frequency in 2001 (the high-water period) varied for the plains from 1% (Belaya–Mishelevka) to 1.5% (Kitoi–Dabady) in the piedmonts of the basins.

The indicators of the relative variability of the size of insular floodplains, which we utilized to evaluate planned deformations, are often minimal, despite the relatively high values of flood frequency indicators. For wide floodplain rivers, the highest values of this indicator in the mountainous and piedmont regions were 0.2–0.7 (i.e., 20–70%

of the floodplain area), while the minimum values in the plains were 0–0.1. Instead of the destruction or alluvium of the banks, the renewal of lateral secondary channels within the flood plains played a major role in the flow–land–flow interaction system. This suggests a relatively high level of channel stability, which is supported by the comparably low rates of bank destruction. As a result, catastrophic floods are more likely to occur. Therefore, in mountainous and piedmont regions, high floods are more likely to result in channel alterations.

The processes of urbanization had a substantial impact on the rate of bank deformations and the intensity of hazardous process manifestations within the plains. It was revealed that the presence of roads within massifs of floodplains frequently caused an increase in flow rates, a rise in the volume of transported material, as well as the development of accumulative landforms. Due to the washing out of small fractions, the soil density in the coastal zone was decreased, which caused landslide deformations and the destruction of infrastructure in urban areas.

The synchronized flooding on the left-bank tributaries of the Angara River has the potential to have a cumulative effect, which will add to the development of hazardous processes and can lead to the destruction of economic objects or cause significant damage to them.

**Author Contributions:** M.Y.O., N.V.K. and A.A.R. conceived the idea and developed the structure of the paper. A.V.S., M.Y.O. and A.A.R. performed channel digitization and mapping. All authors took part in the research on fluvial processes and the preparation of the paper. All authors have read and agreed to the published version of the manuscript.

**Funding:** This work was carried out with the financial support of the Russian Science Foundation (project No. 22-27-00326 "Specifics of the formation and factors of development of river valleys in the basins of the left tributaries of the Angara: modern dynamics and paleogeographic aspects").

**Data Availability Statement:** Not applicable.

**Acknowledgments:** We thank Inna Zlydneva for assistance with the translation of the manuscript and for comments that greatly improved the manuscript.

**Conflicts of Interest:** The authors declare no conflict of interest.

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
