# Peer review of "Channel Deformations and Hazardous Processes of the Left-Bank Tributaries of The Angara River (Eastern Siberia)"

_water, doi:10.3390/w15020291_

Round 1
Reviewer 1 Report
Dear editor:
The paper titled Channel Deformations and Hazardous Processes of the Left- 1
Bank Tributers of The Angara River (Eastern Siberia)” investigated the influence of anthropogenic and natural factors in the trends and mechanisms of development at various topological levels. I think the study topic is interesting and helpful in similar studies. The paper is suggested to be accepted for publication after following modifications:
(1) Adding a figure about the elevation variation along the four rivers after figure 1 may help the readers to easily understand the idea of the paper.
(2) The conclusion part is poorly organized, and main points of the paper are ambiguous.
(3) Too much refences are in Russian, and some of them are too old.
Author Response
Response to Reviewer 1 Comments
Point 1: Adding a figure about the elevation variation along the four rivers after figure 1 may help the readers to easily understand the idea of the paper.
Response 1: Figure about the elevation variation along the four rivers has been added
Point 2: The conclusion part is poorly organized, and main points of the paper are ambiguous.
Response 2: The final part has been rewritten, the conclusions have been improved.
Point 3: Too much refences are in Russian, and some of them are too old.
Response 3: Refences in Russian have been replaced with English ones and new ones have been added. Recent reviews from the Water journal have been added.
We thank the referee for valuable comments aimed at improving our paper.

Reviewer 2 Report
The results seem to be determined on the basis of field observation. Is there a possibility of using any modeling approach to justify the results?
There is a scope for improving the language of the paper. Please improved for grammar.
Most of the reviews are old. It will be better to add a few recent reviews especially from the Water journal
Improve this sentence "They reach 1.5-2 m per year in valleys, and 2-6 m in rivers of mountains and piedmonts" in abstract
Figures do not have axis titles. Please improve the figures and tables.
some of the references do not have the year of publication. Please arrange the references in the style of the journal.
Author Response
Response to Reviewer 2 Comments
Point 1: The results seem to be determined on the basis of field observation. Is there a possibility of using any modeling approach to justify the results?
Response 1: The results are based on the use of field observations along with the use of different estimation methods. A possibility of using any modeling approach to justify the results is not excluded. In this case it was not included in the points of the study. We will consider this
Point 2: There is a scope for improving the language of the paper. Please improved for grammar.
Response 2: We checked and improved the language of the paper and improved for grammar.
Point 3: Most of the reviews are old. It will be better to add a few recent reviews especially from the Water journal
Response 3: Refences in Russian have been replaced with English ones and new ones have been added. Recent reviews from the Water journal have been added.
Point 4: Improve this sentence "They reach 1.5-2 m per year in valleys, and 2-6 m in rivers of mountains and piedmonts" in abstract.
Response 4: This sentence in abstract has been improved: "The highest rate of bank erosion in the plains is 1.5 to 2 m per year; for rivers in mountains and piedmonts, it is 2 to 6 m per year".
Point 5: Figures do not have axis titles. Please improve the figures and tables.
Response 5: Axis titles in Figures have been added. Figures and tables have been revised and improved.
Point 6: Some of the references do not have the year of publication. Please arrange the references in the style of the journal.
Response 6: We checked the references, added the year of publication and arranged the references in accordance with style of the journal.
We express our appreciation to the referee for useful comments aimed at improving our article.
